# In-Season Longitudinal Hydration/Body Cell Mass Ratio Changes in Elite Rugby Players

**DOI:** 10.3390/sports11080142

**Published:** 2023-07-28

**Authors:** Álex Cebrián-Ponce, Cristian Petri, Pascal Izzicupo, Matteo Levi Micheli, Cristina Cortis, Andrea Fusco, Marta Carrasco-Marginet, Gabriele Mascherini

**Affiliations:** 1INEFC-Barcelona Sports Sciences Research Group, Institut Nacional d’Educació Física de Catalunya (INEFC), University of Barcelona (UB), 08038 Barcelona, Spain; acebrian@gencat.cat (Á.C.-P.); mcarrascom@gencat.cat (M.C.-M.); 2Department of Sports and Computer Science, Section of Physical Education and Sports, Universidad Pablo de Olavide, 41013 Seville, Spain; cpet2@alu.upo.es; 3Department of Medicine and Aging Sciences, University “G. D’Annunzio” of Chieti-Pescara, 66100 Chieti, Italy; pascal.izzicupo@unich.it; 4Exercise Science Laboratory Applied to Medicine “Mario Marella”, Department of Experimental and Clinical Medicine, University of Florence, 50134 Florence, Italy; matteo.levimicheli@unifi.it; 5Department of Human Sciences, Society and Health, University of Cassino and Lazio Meridionale, 03043 Cassino, Italy; c.cortis@unicas.it (C.C.); andrea.fusco@unicas.it (A.F.)

**Keywords:** urine, phase angle, bioelectrical, BIVA, team sport

## Abstract

Background: Hydration status has a direct role in sports performance. Bioelectrical Impedance Vector Analysis (BIVA) and Urine Specific Gravity (USG) are commonly used to assess hydration. The study aims to identify the sensitivity and relationship between BIVA and USG in a field sports setting. Methods: BIVA and USG measurements were conducted five times throughout one rugby season. 34 elite male rugby players (25.1 ± 4.4 years; 184.0 ± 7.8 cm; 99.9 ± 13.4 kg) were enrolled. Differences over time were tested using one-way repeated measures ANOVA, and Bonferroni’s post-hoc test was applied in pairwise comparisons. Resistance-reactance graphs and Hotelling’s T2 test were used to characterize the sample and to identify bioelectrical changes. A repeated measures correlation test was conducted for BIVA-USG associations. Results: Two clear trends were seen: (1) from July to September, there was a vector shortening and an increase of the phase angle (*p* < 0.001); and (2) from December to April, there was a vector lengthening and a decrease of the phase angle (*p* < 0.001). USG reported neither changes nor correlation with BIVA longitudinally (*p* > 0.05). Vector variations indicated a body fluid gain (especially in the intracellular compartment) and a body cell mass increase during the preseason, suggesting a physical condition and performance improvement. During the last months of the season, the kinetic was the opposite (fluid loss and decreased body cell mass). Conclusions: Results suggested that BIVA is sensitive to physiological changes and a better option than USG for assessing hydration changes during a rugby sports season.

## 1. Introduction

Rugby is considered an aerobic sport with an intermittent activity profile of frequent changes between high- and low-intensity periods, requiring physical qualities such as endurance, speed, agility, power and flexibility [1,2]. These athletic qualities are influenced by body composition characteristics [3,4]. The body of evidence suggests that as competitive standards rise, rugby players require higher functional body mass (BM) with a lower percentage of body fat [2]. These anthropometric characteristics and athletic aptitudes change across different season periods [5,6]. Therefore, strength and conditioning coaches should consider tracking these parameters to maximize sports performance.

Among the methods developed to assess body composition, bioelectrical impedance analysis (BIA) is an easy and viable technique for obtaining quantitative assessment [7]. However, this approach has some limitations due to the potential inaccuracy of the predictive equation when applied to populations with different characteristics, as well as a minimum standard error of estimation, even when the equation is accurate [8]. In recent years, bioelectrical impedance vector analysis (BIVA) has been introduced within the sports field to investigate body composition and its changes with sport-specific training [9,10,11], overcoming the aforementioned limitation, since BIVA allows the identification of changes in hydration status and mass of soft tissues by solely considering the raw impedance components (resistance, R; reactance, Xc) independently of regression predictions and assumptions of the constant chemical composition of the fat-free body [7,12,13]. The derived impedance (Z) and the phase angle (PhA) can also be obtained from these parameters. According to the classic BIVA approach, Z is inversely related to total body water (TBW) [10]. PhA indicates cellular health and cell membrane integrity, and is inversely related to the extracellular/intracellular water (ECW/ICW) ratio [14]. Therefore, all interpretations should be based on the interpretation of Z and PhA jointly, along with the vector position on the Resistance-Reactance (RXc) graphs [15]. In these graphs, vector displacements parallel to the major axis of tolerance ellipses indicate changes in tissue hydration (vector lengthening indicates fluid loss, whereas a vector shortening indicates fluid gain). Vector shifts parallel to the minor axis of the ellipses indicate changes in cell mass and ECW/ICW ratio: a leftward shift indicates increased cell mass and a decreased ECW/ICW ratio, whereas a rightward shift indicates a decreased cell mass and an increased ECW/ICW ratio [7,12]. Although BIVA has been studied in many sports, the scientific literature on BIVA applied in rugby is scarce, especially at the elite level.

The measurement of urine-specific gravity (USG), which measures the weight of the urine and the particles contained in it [16], is also a commonly used method to assess hydration status among athletes [16,17,18]. However, USG’s day-to-day reliability has not been reported yet [19]. Nevertheless, some research has been made using USG in rugby, detecting that most players are, in principle, hypohydrated [19,20].

As BIVA and USG are standard methods to assess hydration, the present study aimed to examine the sensitivity and relationship between BIVA and USG at multiple time points throughout a competitive rugby season in elite players to measure hydration and body cell mass status. We hypothesized that BIVA and USG methods are sensitive enough to detect the physiological changes in the body in the different stages of the season, with a correlation between them. In addition, we also hypothesized that the bioelectrical change, according to the time of the competitive season, is due to the adaptations of the physical training.

## 2. Materials and Methods

### 2.1. Subjects

This retrospective, quasi-experimental study involved 34 Italian elite rugby players (age: 25.1 ± 4.4 years; height (H): 184.0 ± 7.8 cm; BM: 99.9 ± 13.4 kg; body mass index (BMI): 29.4 ± 3.2 kg/m^2^) from the Rugby Rovigo team who participate in the “Top 10 Championship”. They were all in good health, as assessed by their team physician. No drugs or supplements influencing fluid balance were taken. All athletes participated voluntarily, being informed of the entire process and methodology used, providing their consent in writing in accordance with the Declaration of Helsinki 2013 [21]. The Ethics Committee for Clinical Sport Research of Catalonia approved the study (Ethical Approval Code: 0022/CEICGC/2023). All physical performance data were anonymized before analyses to ensure player confidentiality.

### 2.2. Procedures

Five measurements were performed during an entire season (2014/2015). The first measurement was performed on Jul 22 (T1), the second on Jul 27 (T2), the third on Sep 6 (T3), the fourth on Dec 19 (T4), and the fifth on Apr 18 (T5). All measurements were performed in the same place and conditions (in the morning and after at least 12 h without exercise) and by the same operator (C.P.).

#### 2.2.1. Anthropometric Measurements

BM was measured to the nearest 0.1 kg and H to 0.5 cm using a mechanical column scale model Seca 700^®^ with stadiometer Seca 220^®^ (Seca GmbH & Co., Hamburg, Germany). BMI was calculated as BM/H2 (kg/m^2^).

#### 2.2.2. Bioelectrical Measurements

R and Xc were measured using a BIA 101 Anniversary Sport Edition analyzer (Akern Srl, Florence, Italy) that emitted a 400 mA alternating sinusoidal current at 50 kHz (±0.1%). This device was previously calibrated with a known impedance circuit provided by the manufacturer, whose impedance values were R = 383 ± 10 Ω and Xc = 45 ± 5 Ω.

Subjects were tested according to the manufacturer guidelines, that is to say, with their arms and legs kept from touching the body by non-conductor foam objects to prevent adduction or the crossing of the limbs. Bioelectrical measurements were recorded in the morning before the training session and after at least 12 h without exercise and after a stabilization period of 5 min, in which the players remained laying motionless to ensure the proper distribution of body fluids. The ambient temperature was 22 to 24 °C. Injector electrodes (Biatrodes, Akern, Florence, Italy) were placed on the dorsal surface of the right hand (proximal to the third metacarpal-phalangeal joint) and foot (proximal to the third metatarsal-phalangeal joint). The detector electrodes were placed proximally 5 cm from the injector ones to prevent interaction between electric fields, which could lead to overestimating the impedance values.

Z was calculated as (R2 + Xc2)0.5, and PhA as tan−1 (Xc/R · 180°/π). R, Xc and Z were adjusted by height (R/H, Xc/H, Z/H). R, Xc and Z were adjusted by height (R/H, Xc/H, Z/H). RXc graph was used to plot the five-time points of the rugby players (in confidence ellipses) regarding the 50%, 75% and 95% tolerance ellipses of the reference population, which consists of 139 athletes (age: 21.5 ± 5.0 years; H: 183.3 ± 9.1 cm; BM: 77.2 ± 11.4 kg; BMI: 22.9 ± 2.6 kg/m^2^) of 11 different sports: athletics, basketball, handball, judo, karate, modern pentathlon, rugby, soccer, swimming, triathlon and volleyball [14]. Furthermore, RXc-paired graphs were used to compare bioelectrical differences over time. In these graphs, confidence ellipses overlapping the origin indicate no differences, whereas non-centered confidence ellipses indicate significant changes.

#### 2.2.3. Urine-Specific Gravity Measurements

USG was determined using a refractometer model PEN-Urine S. G. (ATAGO PEN-Urine S.G) [22]. The cut-off criteria for urinary dehydration markers (USG, UOSM) were based on published guidelines [23] and defined as >1.02 (g·mL^−1^) and >1.03 (g·mL^−1^) for hypohydration and severe hypohydration, respectively.

### 2.3. Statistical Analysis

Descriptive analysis was calculated, and data are presented as mean ± standard deviation (SD). After testing each variable for the normality of the distribution (Shapiro–Wilks test), differences in all variables over time were tested using a one-way repeated measures ANOVA. The sphericity assumption was checked using the Mauchly test, and a post-hoc test with Bonferroni correction was applied in pairwise comparisons. An RXc graph was used to characterize the sample. RXc paired graphs and paired one-sample Hotelling’s T2 test were used to identify bioelectrical changes over time. Pearson correlation’s coefficient was used to examine the transversal sensitivity and relationship between BIVA parameters (R/H, Xc/H, Z/H, PhA) and USG in each time point. Repeated measures correlation analysis (rmcorr) [24] was conducted to evaluate intra-individual associations between the same BIVA parameters and USG longitudinally during the season. The relationship was assessed as: small (0–0.3), moderate (0.3–0.5), significant (0.5–0.8) and highly relevant (0.8–1.0). The significance level was set at *p* < 0.05 (two-sided). SPSS (Chicago, IL, USA, ver. 21), RStudio (RStudio Inc., Boston, MA, USA, ver. 2023.03.0) and BIVA software [25] were used for data analysis.

## 3. Results

A total of 34 elite rugby players were measured five times during the season. Each time point is plotted in Figure 1, compared to the athlete’s reference population, including rugby players. Despite the visible differences between the 95% ellipses of time points, they all lie in the lower-left quadrant of the reference tolerance ellipse, mainly within 95% and 50%.

Table 1 shows the BM, bioelectrical and USG changes during the season using five different timepoint measurements. BM progressively increased, reaching its highest point at the end of the season (T1 vs. T5: *p* = 0.022; T3 vs. T5: *p* = 0.021). R/H and Z/H reported the same kinetic, statistically decreasing in T2 (T1 vs. T2: *p* < 0.002) and T3 (T1 vs. T3 and T2 vs. T3: *p* < 0.001). After reaching the lowest values in T3, values increased in T4 (T3 vs. T4: *p* < 0.001), at the baseline (T1 vs. T4: *p* = 1), and also in T5 (T4 vs. T5: *p* < 0.001), well above the baseline (T1 vs. T5; *p* < 0.001). Xc/H follows a similar trend, but being the lowest point at T2, T3 is already increasing, with no differences to baseline (T1 vs. T3: *p* = 1). Both T4 and T5 are significantly higher than T1, T2 and T3 (*p* < 0.001), but there are no differences between T4 and T5 (*p* = 1). PhA remained equal in T2 (*p* = 1), and increased in T3 (T1 vs. T2 and T2 vs. T3: *p* > 0.001). From T3 to T4, there was no increase (*p* = 1). Finally, PhA decreased in T5 (T4 vs. T5, *p* = 0.007) remaining above baseline (T1 vs. T5, *p* = 0.02). USG remained constant throughout the season, with no statistical differences (*p* = 0.754).

In addition to the raw bioelectrical changes, paired graphs (Figure 2) report statistical migration of the complex vector in all time points (*p* > 0.001). Vector shortening suggests a progressive body fluid gain until T3 due to increased ICW, as indicated by the PhA increase. From T3 to T5, a progressive vector lengthening suggests a body fluid loss, peaking at T5. PhA indicated that the ratio ECW/ICW remained constant in T4 but not in T5, due to an increase in ECW or a decrease in ICW.

Players were hypohydrated for almost the whole season, according to Sawka et al. [23], except for T4 when they were euhydrated.

BIVA (R/H and Z/H) and USG were statistically correlated at T5 only (r = 0.43, *p* = 0.019; r = 0.42, *p* = 0.021, respectively). No correlation was detected between any bioelectrical parameter changes for the non-significant USG changes longitudinally throughout the season (Table 2).

## 4. Discussion

This study aimed at assessing elite rugby players’ hydration and body cell mass changes throughout a whole season with two different methods (BIVA and USG) that measure similar aspects, and to check the correlation between these two methods. The main findings of the present investigation were threefold: (1) BIVA identifies hydration/body cell mass changes, while USG is not able to do so; (2) no correlation between BIVA and USG was found; and (3) it is necessary to take into account the temporality of the bioelectrical assessments in the rugby players since the values change significantly.

The rugby players were plotted in the classic tolerance ellipse of an athlete’s population (different sports, including rugby) [14] as the most similar population currently found in the literature (Figure 1). The players’ confidence ellipses were plotted separately according to the assessment time. They all fit within the lower left quadrant, indicating an above-average amount of body water and body cell mass compared to other athletes. These data agree with Campa et al. [26], which indicates that team sport athletes have a greater PhA (and therefore cellular health) than other sports (such as endurance) due to a mesomorphic phenotype and higher muscle mass [27,28]. In addition, the players’ PhA fit within the 50th and >95th percentile of a 147 rugby male sample [26], indicating that the physical condition of our players was above average to their homonyms.

Although all the confidence ellipses’ assessments are within the same quadrant, they are all significantly different from one another (Figure 2). These results align with previous investigations conducted in different sports, reporting that body composition and bioelectrical values change across the competitive season phases [29,30]. During almost nine months of the study, two clear trends can be seen between the different evaluations, especially before and after T3 (the start of the season). From T1 (pre-season) to T3, there was a significant shortening of the vector length (−9.2 ± 4.3%, *p* < 0.001) jointly with an increase in the PhA (9.8 ± 4.8%, *p* < 0.001), indicating a body fluid gain and a decrease of the ECW/ICW ratio. In contrast, from T3 to T5 (end of the season), there was a progressive lengthening of the vector (20.4 ± 7.5%, *p* < 0.001), while PhA remained stable at T4 (2.2 ± 9.5%, *p* = 1) and decreased at T5 (−5.3 ± 7.1%, *p* < 0.001), indicating a body fluid loss and an increase in the ECW/ICW ratio. From T1 to T5, there was a significant lengthening of the vector (9.5 ± 6.9%, *p* < 0.001) jointly with an increase in the PhA (5.4 ± 7.7%, *p* < 0.001), indicating that the players experienced a body fluid loss. However, the ECW/ICW ratio also decreased.

The kinetic of body water content (inversely related to vector length [9]) and ECW/ICW ratio (inversely related to PhA [14]) are important, since it is well reported that these parameters (particularly ICW) play a fundamental role in sports performance, since increases in ICW are associated with power and strength improvements [31,32,33]. Due to the lack of performance data in this study, the knowledge of the changes in ICW, ICW/ECW ratio and PhA are important in order to understand the physical condition of the players throughout the season. In the non-rugby player population, PhA was positively associated with relative power and relative and absolute strength (*p* < 0.05) [34], and vector length shortening was associated with improvements in endurance performance (*p* = 0.034) [29]. Nunes et al. [4] found negative correlations between BIVA and peak running velocity (r^2^ ≥ 0.49), neuromuscular performance tests (vertical jumps, r^2^ ≥ 0.30; and sprinting ability r^2^ ≥ 0.65) and some match performance variables (r^2^ ≥ 0.30) in rugby players. However, these correlations were not based on the raw BIVA values but on the fat mass calculated based on the raw bioelectrical values. These data indicate that our players experienced changes in body composition and physical condition throughout the season that may influence their sports performance [35,36]. The time of the season when players were fittest, according to PhA, was at T3/T4. These data confirm the importance of assessing athletes according to the specific time of the season, as previously reported [29,30].

Alternatively to BIVA, USG has also been widely used to assess hydration status [17]. However, our players not only showed no USG differences throughout the season (*p* = 0.754), but also showed no correlation for any of the bioelectrical parameter changes (*p* > 0.08). Only at T5 was there found to be a transversal moderate positive correlation between R/H and Z/H and USG, which is clearly not enough to state that the two methods are correlated. Marathon runners have also reported this lack of correlation [37]. In contrast, a weakly negative correlation between BIVA and USG has been found in divers, which is the opposite of what the authors expected, since such results indicated that the higher the amount of water, the higher the urine density [38]. In a recent study, Francisco et al. [39] showed that athletes with a low water intake reported higher USG values. However, low and high water intake differences were found in no TBW, ICW, ECW and ECW/ICW ratio (all these parameters were determined using dilution techniques). Based on this discovery, the authors recommend considering USG not as an indicator of hydration status but as a biomarker resulting from those mechanisms to maintain homeostasis. Similarly, Zubac et al. [40] indicated that urinary dehydration markers such as USG seem inappropriate diagnostic tools to screen for fluid deficits in real-life scenarios. Furthermore, it should be noted that athletes with a large muscle mass may present a higher USG without being dehydrated [19] since, according to the cut-off proposed by Sawka et al. [23], our athletes would be hypohydratated during most of the season, contrary to what BIVA indicates, since plotting our subjects in the lower half of the reference tolerance ellipse (Figure 1). These data suggest that if the goal of research or sports teams is to identify changes in the amount of water or its compartments, BIVA is a much more sensible option than USG. It is already well known in the literature that BIVA is a valid, economical and practical method for assessing hydration status and changes in athletes [9,10,11]. However, it can also serve as a method to assess athletes’ cellular quality and physical condition throughout the season. Furthermore, these findings may be engaging in the sports field, since they may help athletes and their physical and technical staff to track the workload and physical condition quickly, easily and economically to improve sports performance. Nonetheless, some limitations must be addressed. First, the most effective strategy for determining hydration is the combination of other methods (plasma osmolarity with the dilution technique) and non-BIVA with USG [41]. However, this combination cannot be used due to the in-field nature of the assessments of the present study. Therefore, interpreting these results relies on methodologies that may not provide information directly. In the case of field settings, finding evaluation solutions that allow the selection of the best possible test battery becomes essential, and the proposal with two practical and transportable methods, such as BIVA and USG, could be a useful solution. Second, the position of the players on the field may influence the body composition (and therefore the bioelectrical values) due to the design training strategies or morphological characteristics. In this study, however, this fact was not considered due to the normality of the sample and the fact that the sample is not excessively large. Third, this study did not collect data on the daily fluid intake of rugby players. However, future studies could predict how the differences found in the present study could change based on the different athletes’ hydration statuses experienced during an entire competitive season.

## 5. Conclusions

The present study suggests that BIVA is sensitive to rugby players’ hydration and body cell mass status, which changes during the regular season. During the pre-season, the players’ physical condition presumably increased and decreased at the end of the season, as vector changes detected it. USG reported no changes, and a moderate correlation with R/H and Z/H only was found in one-time points. This rugby sample presents higher body water content (especially in the intracellular compartment, which predicts strength and power) than other sports. BIVA could thus be a promising tool for monitoring fluid and body cell mass status for rugby coaches and nutritionists.

## Figures and Tables

**Figure 1 sports-11-00142-f001:**
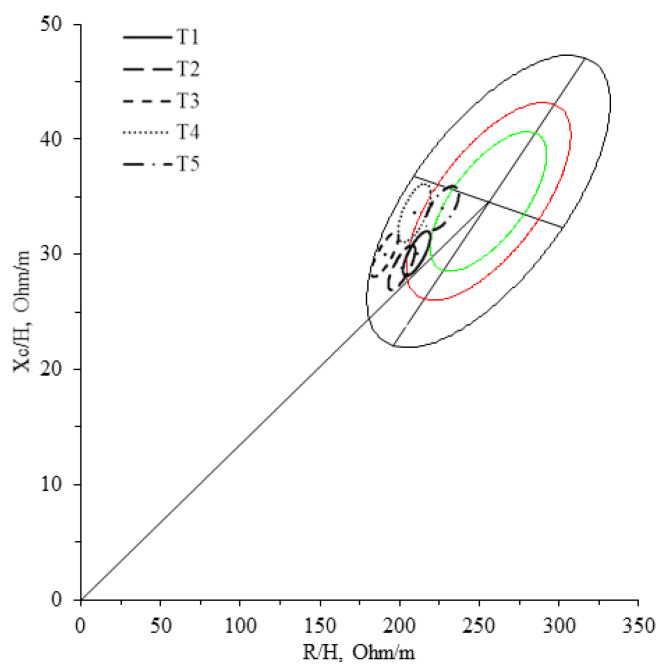
Confidence ellipses of each measurement plotted in the athlete’s reference population [14]. Green line for 50%, red line for 75%, and black line for 95% tolerance.

**Figure 2 sports-11-00142-f002:**
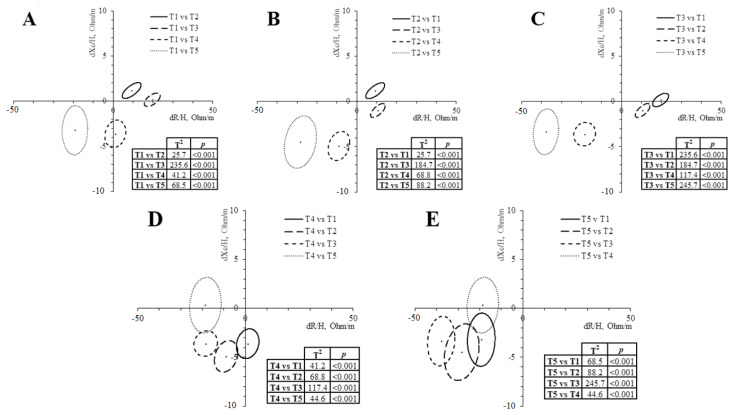
Paired graphs of each measurement. (**A**), differences regarding T1; (**B**), differences regarding T2; (**C**), differences regarding T3; (**D**), differences regarding T4; and (**E**), differences regarding T5.

**Table 1 sports-11-00142-t001:** Longitudinal values through the competitive season.

	T1	T2	T3	T4	T5	ANOVA
BM (kg)	100.7 ± 13.0 ^e^	100.9 ± 13.2	100.9 ± 12.9 ^e^	102.1 ± 13.2	102.3 ± 13.1 ^a,c^	<0.001
R/H (Ω/m)	204.6 ± 18.4 ^b,c,e^	195.2 ± 18.3 ^a,c,d,e^	185.0 ± 18.9 ^a,b,d,e^	205.0 ± 23.0 ^b,c,e^	224.3 ± 24.6 ^a,b,c,d^	0.003
Xc/H (Ω/m)	29.6 ± 4.3 ^b,d,e^	28.3 ± 4.0 ^a,c,d,e^	29.3 ± 4.2 ^b,d,e^	33.3 ± 5.8 ^a,b,c^	34.0 ± 4.5 ^a,b,c^	0.001
Z/H (Ω/m)	206.7 ± 18.7 ^b,c,e^	197.2 ± 18.6 ^a,c,d,e^	187.3 ± 19.2 ^a,b,d,e^	207.8 ± 23.4 ^b,c,e^	226.9 ± 24.8 ^a,b,c,d^	0.004
PhA (°)	8.2 ± 0.8 ^c,d,e^	8.2 ± 0.8 ^c,d,e^	9.0 ± 0.8 ^a,b,e^	9.2 ± 1.2 ^a,b,e^	8.6 ± 0.9 ^a,b,c,d^	<0.001
USG (g·mL^−1^)	1.021 ± 0.009	1.023 ± 0.008	1.023 ± 0.005	1.019 ± 0.009	1.021 ± 0.007	0.754

Legend: BM, body mass; PhA, phase angle; R/H, height-adjusted resistance; USG, urine-specific gravity; Xc/H, height-adjusted reactance; Z/H, height-adjusted impedance.T1, first assessment; T2, second measurement; T3, third measurement; T4, fourth measurement; T5, fifth measurements; ^a^, statistically different from T1; ^b^, statistically different from T2; ^c^, statistically different from T3; ^d^, statistically different from T4; ^e^, statistically different from T5.

**Table 2 sports-11-00142-t002:** Transversal and longitudinal BIVA and USG correlations.

	USG											
	Pearson’s Correlation										Rmcorr	
	T1		T2		T3		T4		T5		Whole Season	
	r	p	r	p	r	p	r	p	r	p	r	p
R/H	0.23	0.216	0.15	0.452	0.07	0.716	−0.08	0.687	0.43	0.019	−0.08	0.438
Xc/H	0.03	0.858	0.01	0.956	−0.05	0.801	−0.15	0.463	0.15	0.428	−0.17	0.082
Z/H	0.23	0.226	0.14	0.462	0.07	0.729	−0.09	0.678	0.42	0.021	−0.08	0.422
PhA	−0.16	0.394	−0.14	0.468	−0.18	0.359	−0.13	0.546	−0.27	0.144	0.15	0.134

## Data Availability

All data generated analyzed during the current study are available from the corresponding author on reasonable request.

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
