# Peer review of "In-Season Longitudinal Hydration/Body Cell Mass Ratio Changes in Elite Rugby Players"

_sports, 2023, doi:10.3390/sports11080142_

Round 1

Reviewer 1 Report

The choice of topic is justified and exciting, the research of biomarkers related to adaptation has scientific and practical benefits.

The BIVA and USG values of 34 elite rugby players were tested 5 times.

They found that measurements showed vector shortening and phase angle increase during the summer period, compared to vector lengthening and phase angle decreases during the winter-spring period. The change in USG was not significant.

Based on the statistical analysis of the results, the authors hypothesize that the relationship between hydration and body cell mass is related to the physical condition of the players.

Even though I find the research valuable, I suggest adding some information to the data:

1.           Has there been data collection on athletes' daily fluid intake?

2.           Is there any information about fluid loss and replacement after trainings and competitions?

3.           Liquid retention is a known fact, whether there was a difference in the actual weather and temperature of the test times.

4.           What information is there about the physical condition of the players, because there is a hint of the decline at the end of the season.

Author Response

The authors would like to thank the reviewer for appreciating our work and suggestions provided to improve our manuscript. Changes made to the manuscript are highlighted in red. Below are the answers to the reviewer’s comments.

The choice of topic is justified and exciting, the research of biomarkers related to adaptation has scientific and practical benefits.

The BIVA and USG values of 34 elite rugby players were tested 5 times.

They found that measurements showed vector shortening and phase angle increase during the summer period, compared to vector lengthening and phase angle decreases during the winter-spring period. The change in USG was not significant.

Based on the statistical analysis of the results, the authors hypothesize that the relationship between hydration and body cell mass is related to the physical condition of the players.

Even though I find the research valuable, I suggest adding some information to the data:

  1. Has there been data collection on athletes' daily fluid intake?
  2. Is there any information about fluid loss and replacement after trainings and competitions?

Answer to question 1 and 2: Thanks for the comment. Data on the daily fluid intake and fluid loss/replacement of rugby players was not collected as it was outside the scope of the study. The comparison between the two methods and the study of the relationship, is unaffected by fluid intake. However, more information has been provided in the limitation of the discussion section to better clarify this aspect, as follows:

“Third, this study did not collect data on the daily fluid intake of rugby players. However, future studies could predict how the differences found in the present study could change based on the different athletes' hydration status experienced during an entire competitive season.”

  1. Liquid retention is a known fact, whether there was a difference in the actual weather and temperature of the test times.

Answer: Thanks for the comment. Assessments were made the morning before the training session. Environmental and seasonal variations are part of the scope of the study, where the different temperatures of the year follow the trend of the study. Furthermore, any differences in ambient temperature recorded compared to the current ones should not have affected the hydration status of the athletes. However, in order to not influence the evaluations, the ambient temperature of the evaluation room was kept constant between 22 and 24°C. This aspect was added in the methods section between lines 117 and 120.

  1. What information is there about the physical condition of the players, because there is a hint of the decline at the end of the season.

Answer: Thanks for the comment. The hint referred to by the reviewer is only a hypothesis that the authors wanted to include in the discussion section based on previous studies of BIVA evaluation during an entire competitive season of other sports (ref. 29, 30).  This speculation is presented within the discussion to enrich the text based on a previous study without, however, having the claim of providing absolute conclusions regarding this aspect (lines 215-217 and 245-246).

We hope that with your suggestions, the manuscript will be more complete and worthy of being accepted for publication.

Thank you very much for your suggestions.

Reviewer 2 Report

Dear authors. I make some suggestions to try to improve your work.

The main limitation of the study is that they do not have a gold standard method to determine the hydration of the rugby players studied (they have indicated it in the text), so the interpretations made of the results may be mere speculations . Please try to justify this aspect.

If your goal is to examine the sensitivity and relationship between BIVA and USG at multiple times throughout a competitive rugby season in elite players to measure hydration and body cell mass status you should compare the results at each time point. these two variables transversally (T1 BIVA vs T1 USG, etc...) and not longitudinal.

After each determination (T1,T2, etc...) and supposedly to detect changes in hydration, indicate what corrections or suggestions were made to the players so that they correct or not their hydration. Indicate how the recommendations could have influenced the results if they were made.

Please remove citation 10 as your topic is myography and not hydration. Please remove citation 29 as it refers to soccer.

Author Response

Dear authors. I make some suggestions to try to improve your work.

The authors would like to thank the reviewer for appreciating our work and suggestions provided to improve our manuscript. Changes made to the manuscript are highlighted in red. Below are the answers to the reviewer’s comments.

The main limitation of the study is that they do not have a gold standard method to determine the hydration of the rugby players studied (they have indicated it in the text), so the interpretations made of the results may be mere speculations. Please try to justify this aspect.

Answer: Thanks for the comment. As we highlighted in the manuscript, this is the main limitation, and we could not control this fact since the data arose from in-field training. In addition, to the authors' knowledge, currently, more than a single marker indicating hydration status is needed. Therefore, two methods were used and compared in a field setting. For better comprehension and to better clarify this fact, we have modified the limitation section, as follows:

“First, the most effective strategy for determining hydration is the combination of other methods (plasma osmolarity with the dilution technique) and non-BIVA with USG [41]. However, this combination cannot be used due to the in-field nature of the assessments of the present study. Therefore, interpreting these results relies on methodologies that may not provide information directly. In the case of field settings, finding evaluation solutions that allow the selection of the best possible test battery becomes essential, and the proposal with two practical and transportable methods, such as BIVA and USG, could be a useful solution.”

If your goal is to examine the sensitivity and relationship between BIVA and USG at multiple times throughout a competitive rugby season in elite players to measure hydration and body cell mass status you should compare the results at each time point. these two variables transversally (T1 BIVA vs T1 USG, etc...) and not longitudinal.

Answer: Thanks for the comment. We appreciate your input and would like to acknowledge that we did indeed calculate the statistic you mentioned. However, upon careful consideration, we found that presenting the interpretation as a single value was more suitable for our study's objectives. Having taken your suggestion into account, we have now incorporated this information in the revised version of the manuscript aiming to provide a more concise and accessible understanding of the relationship between the variables under investigation. Table 2 and a sentence between 250-252 have been added for this purpose. 

After each determination (T1,T2, etc...) and supposedly to detect changes in hydration, indicate what corrections or suggestions were made to the players so that they correct or not their hydration. Indicate how the recommendations could have influenced the results if they were made.

Answer: Thanks for the comment. This is a comment to promote a better nutritional approach to the sports performance of the players. Given the study's retrospective nature, the data were collected in the 2014/2015 season. The team did not have a nutritionist on the medical staff who could directly promote fluid intake. The data collection carried out within this study allowed the Club to pay attention to the role of a professional figure who would allow the nutrition and hydration management of the athletes. Therefore, we are confident to state that no influence/modification was induced by the evaluations throughout the competitive season. However, at the same time, the attention regarding nutritional and hydration aspects was promoted and highlighted.

Please remove citation 10 as your topic is myography and not hydration. Please remove citation 29 as it refers to soccer.

Answer: Thanks for the comment. We have removed citation 10. This reference is a systematic review of the different studies analyzing BIVA at the muscle level (ML-BIVA) in different sports, so we thought that it fitted in the following phrase:

"In recent years, bioelectrical impedance vector analysis (BIVA) has been introduced within the sports field to investigate body composition and its changes with sport-specific training." However, as you indicated, we removed it.

Regarding citation 29, we would like to keep this reference, as in the following paragraphs. We realize that it is focusing on soccer players, however as team sports share some commonalities we believe it is important to highlight in our manuscript that   similar bioelectrical trends in different team sports exist across the competitive season. Furthermore, and as we indicated, a correlation between BIVA and performance tests can be seen in such a study, which is important since we lack this information. 

-line 215-217: "These results align with previous investigations conducted in different sports, reporting that body composition and bioelectrical values change across the competitive season phases [29,30]."

- line 244-246: "The time of the season when players were fittest was at T3/T4. These data confirm the importance of assessing athletes according to the specific time of the season, as previously reported [29,30]."

We hope that with your suggestions, the manuscript will be more complete and worthy of being accepted for publication.

Thank you very much for your suggestions.

Round 2

Reviewer 2 Report

Dear authors: from my point of view, the modifications made improve the understanding of the document. In general, the paper illustrates the use of bioimpedance in the sports environment, although it is necessary to continue advancing in methods that are more direct, efficient and applicable to real sports practice.

I wish you the best